# Psoriasis Management Challenges Regarding Difficult-to-Treat Areas: Therapeutic Decision and Effectiveness

**DOI:** 10.3390/life12122050

**Published:** 2022-12-07

**Authors:** Alin Codrut Nicolescu, Marius-Anton Ionescu, Maria Magdalena Constantin, Ioan Ancuta, Sinziana Ionescu, Elena Niculet, Alin Laurentiu Tatu, Henner Zirpel, Diamant Thaçi

**Affiliations:** 1Medical Center “Roma” for Diagnosis and Treatment, 011773 Bucharest, Romania; 2Dermatology Department, University Hospital “Saint Louis”, University of Paris, 75014 Paris, France; 3Department of Dermatology II, “Carol Davila” University of Medicine and Pharmacy, Colentina Clinical Hospital, 020125 Bucharest, Romania; 4Department of Rheumatology, “Carol Davila” University of Medicine and Pharmacy, “Dr. I. Cantacuzino” Clinical Hospital, 011437 Bucharest, Romania; 5Department of Dermatology III, “Carol Davila” University of Medicine and Pharmacy, 050474 Bucharest, Romania; 6General Surgery and Surgical Oncology Clinic I of the Bucharest Oncology Institute, “Carol Davila” University of Medicine and Pharmacy, 022328 Bucharest, Romania; 7“Prof Dr. Al Trestioreanu” Bucharest Oncology Institute, 022328 București, Romania; 8Department of Morphological and Functional Sciences, Faculty of Medicine and Pharmacy, “Dunarea de Jos” University of Medicine and Pharmacy, 800008 Galati, Romania; 9Pathology Department, “Sfantul Apostol Andrei” Emergency Clinical Hospital, 800578 Galati, Romania; 10Clinical Medical Department, Faculty of Medicine and Pharmacy, “Dunarea de Jos” University of Medicine and Pharmacy, 800008 Galati, Romania; 11Dermatology Department, “Sfanta Cuvioasa Parascheva” Hospital of Infectious Diseases, 800179 Galati, Romania; 12Research Institute and Comprehensive Center for Inflammation Medicine, University of Lübeck, 23538 Lübeck, Germany; 13Comprehensive Center for Inflammation Medicine, University of Lübeck, 23538 Lübeck, Germany

**Keywords:** psoriasis, psoriasis scores, difficult-to-treat areas

## Abstract

Psoriasis is not optimally controlled in spite of newly developed treatments, possibly due to the difficulty of objectively quantifying the disease’s severity, considering the limitations of the clinical scores used in clinical practice. A major challenge addresses difficult-to-treat areas, especially in the absence of significant body surface involvement. It is controversial whether the severity evaluation of patients with several affected areas (having at least one difficult-to-treat area) should be done differently from current methods. Scores used for special areas (PSSI, NAPSI and ESIF) allow an accurate assessment of disease severity in difficult-to-treat areas, but the issue of whether to integrate these scores into PASI, BSA or DLQI remains. The review’s purpose resides in providing an overview of the main current issues in determining psoriasis severity in patients with psoriasis in difficult-to-treat areas and suggesting possible solutions for the optimal integration of the area assessment in current scores: severity can be either established according to the highest calculated score (PASI or PSSI or NAPSI or ESIF) or by adding a correction factor in the calculation of PASI for special areas.

## 1. Introduction

Psoriasis, an autoimmune systemic skin disease, has joint involvement in 30% of cases (psoriatic arthritis—PsA) [1,2]. Psoriasis management brings many challenges, including an increased prevalence over the past years, chronicity, high grade of disability for some cases and associated comorbidities [3]. Psoriasis plaques are more common on the extensor surfaces but may be present in any area of the body, including the scalp, groins, and genital area. Nails are frequently affected and can be seen as isolated locations [1,2]. Genetic factors are various [1,2,3], and other factors that may trigger or exacerbate psoriasis can include: stress, body mass index (BMI), infection, drugs (beta-blockers, lithium, angiotensin-converting enzyme (ACE)-inhibitors, (synthetic) antimalarials, tetracyclines, non-steroidal anti-inflammatory drugs (NSAIDs)), withdrawal of systemic (or even potent local) corticosteroids, chronic alcohol consumption, smoking, friction, minimal trauma of the skin—even a slight irritation, radiotherapy, endocrine disorders, and many other [3,4,5,6].

Psoriasis is a persistent public health problem involving almost 125 million humans all around the globe [7], although 81% of the world’s countries are lacking in psoriasis epidemiology [8]. The prevalence within adult populations estimates at between 0.91% in the United States of America (U.S.A.) and 8.5% in Norway, increasing in the world from 758/100,000 cases in 1990 to 812/100,000 cases in 2017. The highest incidence rates are in the European region, where an increase in incidence from 143.7 cases per 100,000 in 1990 to 147.2 in 2017 was reported [4,9,10]. This supports the statement that there is great variation in the prevalence rates among the world’s regions, with variations from 0.73% (to even 2.9%) in Europe or 0.7 (even 2.6%) in the United States, to under 0.5% in Asia (China, Sri Lanka), Latin America, India, or Africa (Tanzania, Egypt) [11].

In Romania, psoriasis affects approximately 400,000 people, as evidenced by the first epidemiological study published recently in 2021 [12]. The prevalence of psoriasis vulgaris within this study was 4.99% [12] (although there are other sources reporting percentages as high as 5.18) [13]; this study is currently continuing in order to assess factors contributing to the increase in psoriasis prevalence over time [12].

Psoriasis displays diversity in presentation and treatment results, varying with disease extent in time, affected areas and affected body surface percentage. Perception of psoriasis severity differs at baseline from long-term disease course. Disease severity guides treatment decisions, choice of medication and the intensity of the treatment response, as well as eligibility criteria for participation in phase II or III clinical trials. In daily clinical practice, physicians evaluate psoriasis severity by combining subjective and objective parameters involving the skin involvement extension, signs and symptoms, and also the specific location of lesions and the impact on every patient’s quality of life [14,15].

There is great variability in quantifying the severity of the disease worldwide, and we aimed to evaluate the current guidelines/recommendations and the severity scores in order to achieve better results as an integrated approach to the disease by combining the existing scores in current therapeutic guidelines, recommendations and protocols.

## 2. Materials and Methods

The current work is the outcome of conducting a thorough, comprehensive analysis of the current specialty literature, generating a narrative review type of article by using updated materials starting from the year 2000 and beyond, with the exclusion of prior materials and data reported in languages other than English. Database/record searches were carried out using Pubmed, Web of Science, and Google Scholar, by utilizing such keywords as: “psoriasis”, “psoriasis scores”, “PASI”, “BSA”, “PGA”, “IGA”, “DLQI”, “PSSI”, “NAPSI”, “ESIF” and “difficult-to-treat areas”, individually or combined, generating the current work after selection of the most representative, relevant and work-related, pertinent articles. After searching three databases and zero registers, eliminating articles irrelevant to the current subject, based on 59 relevant results, the following review was compiled.

## 3. Current Challenges in Disease Severity and Treatment Goals

In the past decade, new treatments and treatment strategies have become available, especially for those that suffer from moderate-to-severe forms of the disease. Amidst these newly developed drugs, biologics ensure the selective immune-mediated pathway inhibition involving cytokines: tumor necrosis factor (TNF), interleukin 23 (IL-23), IL-17, IL-36, etc. [16,17]. Besides all of these advancements, psoriasis is still not always optimally treated; the patient satisfaction rate with the existing therapies remains modest; meanwhile, the disease burden is at high levels, in spite of the effectiveness of new treatments [17,18].

An explanation for the suboptimal response to treatment and outcomes could be the difficulty in quantifying the disease severity and the limitations brought by the commonly used clinical scores. Quantifying disease severity has implications for treatment selection and assessing therapeutic efficacy. There already exist many severity classification systems, but without reaching a general agreement or allowing the explicit separation into the moderate and the severe disease types, although making this clear-cut dissociation could make a difference in making the therapeutic choice. Reaching a consensus on setting a clear definition for severity is of special importance in medical research, clinical practice, and also for insurance and healthcare authorities in order to evaluate the extensive range of therapies now available for psoriasis [19,20,21,22].

Psoriasis severity includes the subjective or objective assessment of the disease: physical aspects, symptoms, the disease impact and the long-term disease and treatment response history categorization. There are numerous methods for assessing the severity of the disease: Body Surface Area (BSA) involved, Psoriasis Area and Severity Index (PASI), Physician’s Global Assessment (PGA), Investigator Global Assessment (IGA), Dermatology Life Quality Index (DLQI). There is great variability in quantifying the severity of the disease worldwide; the PASI score is frequently used in Europe, both for assessing the disease severity and for treatment response monitoring. PASI is less frequently used in daily clinical practice in the USA, for example, where dermatologists make use of the BSA and IGA in their evaluation for making a therapeutic decision [19].

### 3.1. Body Surface Area (BSA)

BSA is a severity assessment tool that implies the estimation of the extent of body surface involvement by measuring the total area of the body affected by psoriasis. The “palm method” takes into account that the palm of the patient is the equivalent of 1% of the body surface (total BSA = 100% equivalent to 100 palms). Psoriasis may be considered: mild if <3% BSA, moderate 3% to 10% BSA, and severe > 10% BSA [19].

When using the percentage of BSA as an indicator of psoriasis severity, it is important that the measurement be made as accurate as possible. There are many question marks regarding the accuracy of BSA: does ‘the palm’ imply the actual palm or the palm’s entirety, with the fingers and thumb? The full stretched hand area, with the digits, is almost 0.8% for males and 0.7% for females. Assuming incorrectly that the palm area is 1% BSA might lead to an almost 50% misrepresentation of the measurement. According to the recommendations, the entire surface of the palm with five digits is roughly 1% [23].

### 3.2. Physician‘s Global Assessment (PGA)

The PGA is another tool used in order to assess the severity of psoriasis, which uses a scale with 7 items ranging from clear to severe. Global assessments are used in evaluating extensive disease forms and also for localized plaques. Two disease forms are considered—the static one, measuring the doctor’s disease assessment at a single point in the patient’s disease course, and a dynamic one evaluating the global amelioration starting from the baseline. Starting from the idea that it is difficult for physicians to remember the severity of psoriasis at the time of initial diagnosis or in subsequent monitoring, the static PGA has become the standard option for practice use [24].

### 3.3. The Psoriasis Area and Severity Index (PASI)

PASI represents an extensively utilized tool in psoriasis trials that assesses and sets into grade the lesions’ severity and the treatment response. This score evaluates the scaling, redness, induration, and extent of plaques per body region.

The PASI is limited in accuracy establishing. The first limitation comes from the specific PASI formula and location of lesions for hard-to-treat psoriasis. Considering the case of a patient with 3 affected areas, with minimal trunk involvement and a moderate-severe involvement of the scalp and/or face—the severity of the score using PASI could be deficient and lead to an under-treatment of the patient.

The second limitation is the estimation of the percentage of BSA extension—in which case, there is much variation in those suffering from limited lesions, resulting in poor detection of changes in mild or moderate psoriasis. The third limitation refers to patients having variable manifestations of psoriasis with identical PASI scores; for example, one has widespread but mild severity psoriasis lesions, and another has localized, severe lesions. Plaque elevation, erythema and scaling are scored equally, treatments improving erythema or scaling will change the score even more than equally useful treatments that do not—the latter being the fourth limitation. Absolute PASI is recommended as being a more accurate measurement for daily practice (PASI < 1 (minimal), PASI < 3 (very mild), PASI < 5 (mild), PASI 5–10 (moderate) and PASI > 10 (severe)) [25].

### 3.4. The Investigator Global Assessment (IGA) Scale

The IGA scale is frequently used as the final point to reach in psoriasis treatments [23,24,25]. The IGA scale is a visual assessment tool that consists of a score ranging from 0 (clear) to 4 (severe). For a treatment to be considered successful, the affected area must receive a score of 0 or 1 and experience a two-point improvement from the base line. The IGA scale is a visible evaluation device that consists of a rating ranging from zero (clear) to four (severe). The skin rated 4 is bright red, markedly elevated, with a thick, non-tenacious scale. A successful therapy considers that the area involved needs to have a score of 0/1, with a 2-point improvement from baseline.

### 3.5. The Dermatology Life Quality Index (DLQI)

The DLQI is a questionnaire reported by patients which were developed because the patient’s life quality is crucial in proving that skin lesion severity has an effect on the latter. The questions that make up the DLQI are scored on a four-point scale, as follows: 0, not at all/not relevant; 1, a little; 2, a lot; and 3, very much, giving a total DLQI score from 0 to even 30 [26,27,28].

The DLQI assesses principally physical limitations, and few questions address the skin diseases’ psychological involvement, aiming at decreased conceptual validity, specifically in psoriatic disease. Clinical practice proves that DLQI is a superior assessment tool for severe acute skin diseases and less for those with mild impact or few physical signs but with important psychological impact (vitiligo, basal cell carcinoma, alopecia areata). Another concern related to DLQI is that “not relevant” responses could be interpreted as “not at all” answers; this may lead to bias which could undervalue the severity of the disease. Recent studies highlighted the fact that the patients who responded as “not relevant” had a higher degree of severity than those who responded as “not at all”. The rate of “not relevant” answers was high in moderate-to-severe disease patients supporting the statement that those using “not relevant” to one or more questions are going to indicate higher severity of the disease. Because a DLQI score of ≥10 is used for defining severe psoriasis and may influence the decision to initiate biological treatment, this underestimation of disease severity for patients with one or more “not relevant” answers could limit their access to more advanced treatments [26,27,28].

Treatment goals are a way of guiding physicians toward providing the best possible outcomes for patients so that, in the end, they can optimize patient care. Multiple definitions of psoriasis severity exist, which include disease classification mixtures of assessor- and patient-reported measures. The rule of “tens” is well known, and it describes a ‘‘severe psoriasis’’ if the BSA involved is >10% and/or PASI > 10 and/or DLQI > 10. There is a high degree of variability concerning the clinical guidelines establishing the treatment goal and the success or failure of the treatment (Table 1). At the beginning of 2021, a consensus was reached regarding (1) the clear separation between the best possible goals and those that are realistic in setting the therapeutic aim in psoriasis (moderate to severe); (2) the regulation of treatment aims needs to be adjusted indifferent to the existing treatment on the market; (3) the definition of treatment non-performance/inadequacy when not achieving PASI75; (4) that absolute PASI is to be favored to PASI improvement from what is considered baseline; (5) that treatment aims could be influenced by disorder attributes, the diseased’s needs and doctors’ judgment, as well as adherence [29,30].

We consider it important to continuously monitor the treatment effects and make necessary changes during maintenance therapy. Treatment aims can be set for all patients, but treatment changes/adaptations need to be adapted to each patient when the aim is not attained [31].

### 3.6. The Psoriasis Scalp Severity Index (PSSI)

The PSSI measures the psoriasis skin extension and the severity of the scalp erythema, desquamation and infiltration. The psoriasis-affected skin and the degree of severity for the PSSI are established by doctors, which ranges from 0 to 72, 0 meaning absence of psoriasis, with increasing scores revealing more severe disease [32,33,34,35,36,37,38,39]. The PSSI excludes the face and neck areas. The scalp is a prevalent difficult-to-treat area for psoriasis; the visibility of lesions and pruritus degree that are associated with scalp lesions may adversely affect the quality of life of the patient.

### 3.7. The Nail Psoriasis Severity Index (NAPSI)

The NAPSI is a nail psoriasis objective assessment instrument that is used to assess the severity of nail bed and matrix psoriasis by the area of involvement of the nail unit (each nail is divided into 4 quadrants and given a score for nail bed psoriasis and nail matrix psoriasis, depending on the presence of any of the features of nail psoriasis in that quadrant–leukonychia, pitting, red spots in the lunula, crumbling (0 for none, 4 if present in 4 quadrants of the nail) and respectively, onycholysis, splinter hemorrhages, subungual hyperkeratosis, “oil drop” (0 for none, 4 for 4 quadrants)). Each nail gets a matrix score and a nail bed score, the total of which represents the score for that nail (0–8); the sum of all the nails’ scores is the total NAPSI score (0–80 or 0–160 if the toenails are included in the calculation) [32].

Nail damage occurs in only 1 to 5% of patients, with 1 in 2 patients with psoriasis being affected by nail psoriasis at any given time; an estimated lifetime incidence of nail damage ranges between 80 to 90%.

### 3.8. The Erythema, Scaling, Induration, and Fissuring (ESIF)

ESIF is assessed for palm, and sole psoriasis using a 4-point scale (from 0 = clear to 3 = severe) and is determined with the addition of scores for the 4 sole signs, with a total from 0 (absence of disease)–24 (most severe involvement) [37].

A cross-sectional study that included more than 4000 adults with psoriasis from the Danish Skin Cohort evaluated the involvement of hard-to-treat areas. The most frequently difficult-to-treat areas are the scalp (in 43.0% of patients), the face (29.9%), nails (24.5%), soles (15.6%), genitals (14.1%), and palms (13.7%). Sixty-four point 8 percent, 42.4% and 21.9% of patients had involvement of at least one, and respectively at least 2 and at least three difficult-to-treat areas. According to its severity (Table 2), the prevalence of psoriasis in the face, scalp, nails genitals was directly proportional to the severity of psoriasis; for example, 66.1% of patients with severe psoriasis have lesions on the scalp. Among patients with mild psoriasis, 80.4% had an involvement ≥1 difficult-to-treat area, and the frequency among those with severe forms increased to 89.0%. Sixty-eight point eight percent and 43.7% of patients suffering from severe psoriasis had at least 2 and, respectively, 3 difficult-to-treat areas. Courses of action based on results among those with difficult-to-treat areas suggest that they are the population with the highest disease burden [38].

These difficult-to-treat areas have a limited degree of response to local treatment and could be classified as moderate or severe psoriasis, even when BSA ≤ 10 and the PASI ≤ 10. Data collected from the Corrona Psoriasis Registry revealed that 2/3 of psoriasis patients undergoing biological treatment have psoriatic arthritis and/or at least one form of psoriasis in an area difficult to treat (scalp, nail psoriasis, palmoplantar). Scores dedicated to these special areas (PSSI—Psoriasis Scalp Severity Index, NAPSI, ESIF) allow a precise calculation of disease severity, but they are not integrated into the more commonly used scores such as BSA, PGA, or PASI [39].

The biggest challenge in establishing the disease severity, therapeutic choice and efficiency monitoring are “difficult-to-treat” areas, especially in the absence of significant involvement of the body surface elsewhere. The location and morphological features of scalp, nail, palmoplantar and genital psoriasis can often lead to ineffective topical treatment and often requires systemic treatment [37,40,41,42].

Determining psoriasis severity for those suffering from difficult-to-treat areas of psoriasis can be a demanding task due to the fact that some severity definitions depend on the involved area ratio. Dedicated scores allow a more accurate calculation of disease severity in special areas, but the dilemma remains whether or not to assess the overall severity of the disease by integrating these special area scores into scores such as PASI, PGA, BSA and DLQI. It is debatable whether, in patients with several affected areas (a common situation in clinical practice, with at least one special area), the classification of the degree of severity should be made differently from current methods or not.

We have an example of a psoriasis patient having 2 involved skin areas: scalp and trunk (Table 3). The scalp has erythema, induration and exfoliation of over 70%. Trunk lesions extend up to 10% with a severity of 4. The PASI score for this patient is 8, indicating a moderate form of the disease; if in this patient’s case, we would use the PSSI score, the value will add up to 60 points, indicating a severe form, according to European guidelines. Severity framing can significantly influence therapeutic options and the choice of effective treatment for both areas involved [43].

In a series of 26 psoriasis patients having nail psoriasis [44], there was an important(moderate) positive NAPSI and DLQI correlation (*p* = 0.001). A meta-analysis published in 2019 showed that the frequency of palmoplantar psoriasis (PPP) of both palms and soles (59%) was almost 3 times higher than the frequency of any single PPP location, be it palms (21%)/soles (20%). More than 60% of the patients from the 15 studies included in this meta-analysis had PPP with at least one additional area involved [45,46].

The severity assessment in patients with several affected areas and at least one difficult-to-treat area can also raise issues from the point of view of insurance and health authorities: the existence of country-specific therapeutic protocols limits the use of systemic therapy (biologics, small molecules) for severe psoriasis only to be done according to the PASI/PGA/BSA/DLQI scores. To meet psoriasis patients’ needs while respecting the requirements of the authorities involved, we propose to be taken into account the score that represents the highest degree of severity.

The possible solutions for such practical issues reside in making the classification by the degree of severity according to the highest calculated score, regardless of whether it is PASI or PSSI, NAPSI, or ESIF. If, for the 3 examples mentioned above (affecting at least 2 areas, one of which being difficult-to-treat), the severity assessment would be done according to the specific scores of the special areas, then the patients would fall into the category of severe psoriasis, with all of the implications deriving from this classification (choosing the right treatment according to the involvement of a difficult-to-treat area, setting the therapeutic goal, evaluating the success or failure of the therapy). The challenge of this new possible classification could be the subsequent monitoring of the effectiveness of the treatment.

Since PASI is considered one of the standards in patient trials (with treatment comparison, having good correlation with multiple objective outcome measures, being the most validated objective measurement of psoriasis severity, with a test-retest variability of less than 2%), another solution could be to add a correction factor in the calculation of PASI for special areas, as in the example of the first patient with 2 areas involved, scalp and trunk (Table 3). The introduction of a correction factor for that area of the body defined as hard-to-treat could change the importance of the scalp involvement in the calculation of PASI and, consequently, the patient’s framing in the severity degrees, reflecting the reality of usual clinical practice.

## 4. The Challenges of Individualized Treatment and Evaluating the Treatment Success

The issue of psoriasis in “difficult-to-treat” areas—Psoriasis localized in special areas—is difficult to treat and is often associated with important physical disability and discomfort. “Difficult-to-treat” areas are used for describing psoriasis located on the scalp, palms and soles, and nail, being frequently connected to high emotional and functional impact [32,33,34,35,36,37]. Some authors also include as part of this classification the psoriasis of the face and inverse psoriasis [37].

The biggest challenge after establishing the severity of the disease is choosing the optimal therapy adapted to the patient’s needs, especially in cases with multiple site involvement and in difficult-to-treat areas [47,48].

Recent advances in systemic treatments may lead to more benefits in patients for whom topical approaches pose challenges. Scarce-controlled trials had results onsystemic therapy efficacy and safety (traditional or biological) for psoriasis management in difficult-to-treat areas. In general, the available evidence is obtained from sub-analyses of trials which included patients suffering from psoriasis and/or PsA, also having an assessment of the involvement of the nails, scalp, palms or soles [48].

Another reason for poorer clinical outcomes could be related to the patient. Conventional systemic therapies need various concentrations for local treatment response, toxicity being an important problem. In addition, selecting the appropriate treatment is problematic due to the lack of clinical trial data in such locations. Only patients with a minimum of 10% BSA are allowed in the clinical trials for newer targeted biological agents, and that’s why extended evidence regarding the efficacy of new treatments in patients with lower BSA involvement or disease involving difficult-to-treat areas is lacking. This approach impacts the access of patients with decreased disease severity to new therapies in some national health systems and other payers refusing reimbursements for those lacking a BSA involvement of at least 10% [48,49].

## 5. Treatment Goals and Treat-to-Target

In order to reduce the risk of severe comorbidities, doctors should aim at establishing the goals of treatment so that treatment optimization is achieved, along with the long-term quality of life. Treatment goals need to be accustomed to disease severity and possible improvement degree. However, personal treatment aims can vary significantly, surprisingly, between patients with similar disease severity patients. Treatment goals need a clear discussion with the patient at initiation so that to meet both patient and physician expectations. Once target goals are defined, an aimed treatment manner can be contrasted with standard care in order to find the optimal management of the disease (Table 4) [47].

The introduction of biotherapies was accompanied by an increase in the therapeutic target, the PASI 75 response being replaced today by PASI 90 and PASI 100 with the new therapeutic classes. These types of aims may need reconsideration in those with lesions localized in difficult-to-treat areas (scalp, face, soles, palms, nails, genitals), having negative emotional effects, with increased disease severity (as compared to the disease severity evaluated with objective measures (like BSA or PASI) [50,51].

“Treat-to-target” is a concept that describes the changes in treatment as being suitable until a designated aim has been attained. The aimed treatment strategy may also be tough to enforce in world scientific medical practices if the goal purpose is too ambitious and past the attain of the majority of sufferers. PASI 100 needs to be an ideal aim, as only a few present treatments are able to reach it in over 50% of sufferers. The UNCOVER-1 study (Study in Participants With Moderate to Severe Psoriasis) revealed that 35.3% of ixekizumab-treated patients had PASI 100 in the 12th week. Conversely, PASI 90 is viable in many psoriasis patients (70–80%) medicated with IL-17 and IL-23 inhibitors [52,53,54].

Other treatment aims’ analyses could be useful in setting a link between various PASI results. Data from clinical studies on secukinumab showed that achieving a PASI 90 was linked to an improved life quality (DLQI 0/1) in the 12th week, as compared to PASI 75–89. When defining treatment goals, it is necessary to accept the fact that a decrease in PASI is not relevant for all patients. As a result indicator, instead, the modification of absolute PASI might be even more pertinent. According to the Spanish Psoriasis Group consensus, absolute PASI is considered to be useful for medical care by superiorly correlating with DLQI, as opposed to relative PASI amelioration. A reduction of the biological therapy dosage might be attainable in those having complete/near complete results (PGA 0 or 1; PASI 90; absolute PASI from <2 to 3). The criteria necessary in order to go back to biological therapy (in full dosage) include that the absolute PASI values be ≥5 or that there be an absence of a PASI 75 response [55,56].

The European consensus on the treatment aim for psoriasis reveals the necessity of minimal improvement degree and treatment-specific assessment time points, but not all countries mention these minimum evaluation criteria in their guidelines.

## 6. Treatment Success or Failure for Patients with Difficult-to-Treat Areas

There is treatment failure in accordance with the overall treatment guidelines/consensus records—what was relevant was the absence of improvement in PASI 50 during the induction and maintenance phases. These concepts/theories adopted decreased improvement levels if there was good quality of life, with ≥2 taking into account other factors such as patient preference and treatment adherence.

One of the questions that have risen in recent years has been related to the implications of setting higher treatment targets, driven by the very good results of newer biologics [56]. Unfortunately, new biologic therapies are still restrictedly used in a small number of patients (mainly in those suffering from severe psoriasis). For many patients, there is the risk that failure in achieving the goals will be defined as treatment failure and therefore lead to an unnecessary change in therapy. We consider that a clear distinction is needed between the therapeutic goal (for which the doctor and the patient aim) and the minimum response criterion (which regulates the change in treatment).

The minimum response criteria are defined as a PASI 50 response and a DLQI score of 5 units lower than the baseline, assessed every 6 months; failure of treatment means that there is a lack of PASI 50 improvement. At the same time, if there exists an amelioration in PASI 50witha DLQI of not less than 5 units from the beginning, then this situation is also considered treatment failure (Figure 1). The long-term treatment goal is similar to the recommendations from the EuroGuiDerm Guideline on the systemic treatment of Psoriasis vulgaris: at least PASI 90 and DLQI < 2. We consider that this approach in setting the therapeutic target takes into consideration clinical situations with patients having difficult-to-treat areas or patients with multiple areas, including at least one difficult-to-treat area while maintaining an equally ambitious therapeutic target [57].

## 7. Discussion

Psoriasis found on the face, scalp, intertriginous areas, hands, feet, nails and genitals are often diagnosed poorly and under-treated. In spite of the small surface area, which is commonly affected by psoriatic lesions in such areas, patients have increased physical impairment and emotional distress. Limitations in current disease severity scores do not fully assess the impact of disease on the quality of life of patients, and many are not receiving adequate care. In these cases, the therapeutic attitude to adapt is to adapt therapy (dose increase, therapeutic combination), change of treatment (switch) or continuation of treatment for the next 3 months with reevaluation [58]. Psoriasis severity classification is a major problem that needs to be addressed in order to guide the physician’s decisions regarding treatment, always having in mind that the disease is heterogeneous in clinical expression and response to treatment, in its duration and involved areas (including the percentage of body area), being constantly variable [22].

Psoriasis is highly influenced by external factors; the list of triggering factors for flare-ups is extensive, starting with stress, mild localized skin trauma, different infections and drugs for different comorbidities, alcohol consumption, smoking, weather etc. A total of 73% of the patients have at least one comorbidity that can influence the disease evolution and response to treatment, especially for difficult-to-treat areas [58]. In the given example, the patient with 2 areas affected by psoriasis has been evaluated after 3 months of treatment with ∆PASI 90 for lesions on the trunk and ∆PASI < 50 for the scalp. The patient suffered a great level of stress during these 3 months, but according to aDLQI, there was a significant improvement.

There are many systems classifying psoriasis severity, but none have managed to reach a consensus, not having clear-cut demarcation between severity degrees (moderate and severe), the methodology disregarding the involvement of psoriasis in difficult-to-treat areas; the fact that current practices employ systems that are mainly used by physicians/dermatologists (such as PASI, PGA and/or BSI) and by patients (DLQI) alike is a step further in psoriasis severity assessment and treatment, the quality of life being of paramount importance [22]. Figure 2 reveals the raised issues, with point-by-point explanations and proposed solutions for the clinicians in their daily practice.

For psoriasis involving difficult-to-treat areas, the challenge extends from establishing severity and choosing the optimal treatment to correctly evaluating the efficacy or failure of the therapy; this is a major problem that our paper addressed, having the advantage of bringing forth a practical, clinical issue which dermatologists face. There are several solutions to this issue; the first one is the evaluation of different scores for each area and the success or failure of the treatment to be evaluated by the area with the smallest improvement. The second solution is the inclusion of special areas in the treatment goal algorithm or to use of a completely different algorithm for psoriasis in difficult-to-treat areas by using the data obtained in different clinical studies or real-world experience. The current paper has made a summary of the literature at hand, examining the current knowledge on psoriasis and its score assessment, with the limitation of possibly having a broad area of research data (a wide range of psoriasis score reporting analysis), but with the advantage of identifying possible new research areas.

## 8. Conclusions

In clinical practice, psoriasis severity is usually categorized as “mild”, “moderate”, and “severe”, using measurement tools that underestimate the actual disease severity in cases where skin lesions involve ‘special areas such as the face, palms, soles, genitalia and scalp. It is necessary to consider the lesions’ location and patients’ quality of life in order to assess psoriasis severity more fully and accurately. The solutions considered for evaluating difficult-to-treat areas reside in establishing the severity according to the highest calculated score (PASI or PSSI, NAPSI, or ESIF) and/or in the addition of a correction factor when calculating PASI for special areas. The algorithms used to monitor and evaluate therapeutic efficacy should be upgraded with emphasis on the difficult-to-treat areas which greatly influence the overall severity degree.

The need to redefine disease severity with the inclusion of special areas is becoming increasingly obvious and should be done practically in order to prove its usefulness in clinics and research.

## Figures and Tables

**Figure 1 life-12-02050-f001:**
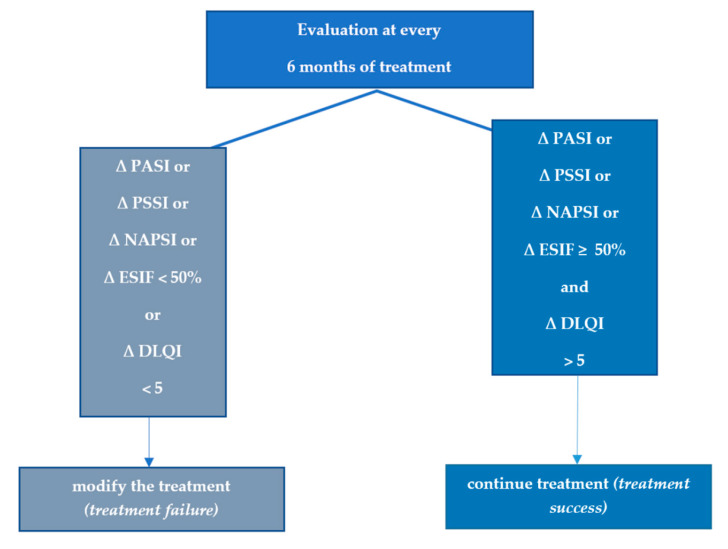
Minimum efficiency criteria for moderate-to-severe psoriasis; ∆ = improvement of the score compared to baseline. Abbreviations: PASI—Psoriasis Area and Severity Index; BSA—Body Surface Area; PSSI—Psoriasis Scalp Severity Index; NAPSI—Nail Psoriasis Severity Index; ESIF—Erythema, Scaling, Induration, Fissuring.

**Figure 2 life-12-02050-f002:**
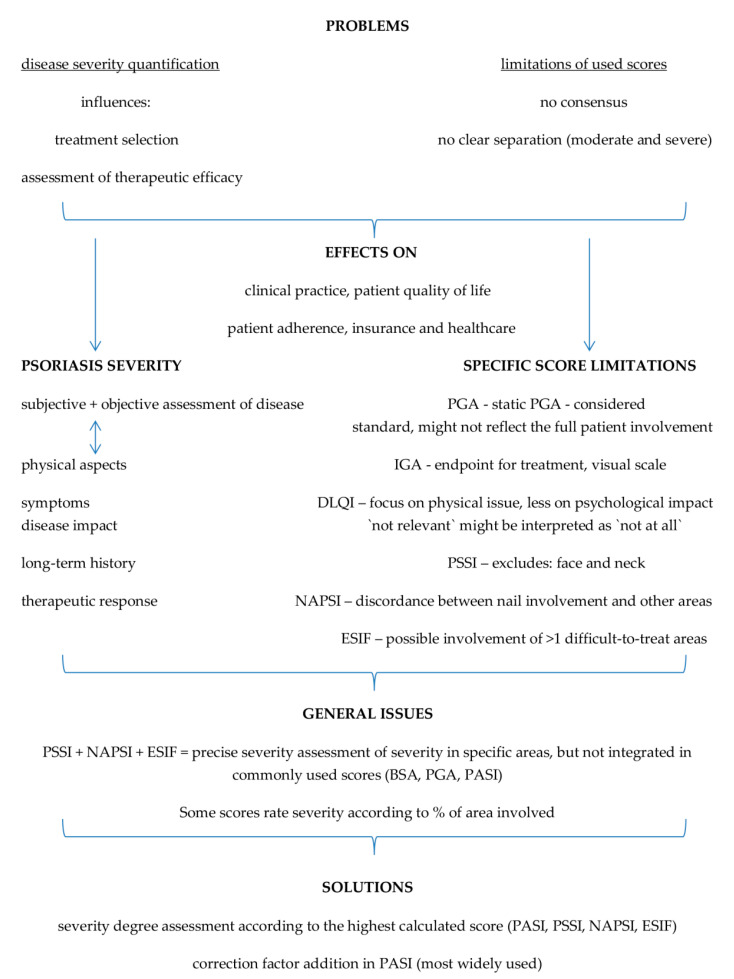
Issues and possible solutions in psoriasis severity effective calculation. Abbreviations: PASI—Psoriasis Area and Severity Index; BSA—Body Surface Area; PSSI—Psoriasis Scalp Severity Index; NAPSI—Nail Psoriasis Severity Index; ESIF—Erythema, Scaling, Induration, Fissuring.

**Table 1 life-12-02050-t001:** Treatment goals and response and/or failure definitions, with treatment changes in those with moderate to severe psoriasis [29,30,31].

Guideline	Moderate to Severe Psoriasis Treatment Aim Definitions	Treatment Response and/or Failure Definitions, with Changes in Those Suffering from Moderate to Severe Psoriasis
European guideline of systemic therapy	Any psoriasis treatment should aim at eliminating all symptoms of skin inflammation.Necessary: minimum improvement and particular drug analysis times.	Throughout the phases of induction and maintenance: ○ PASI 75—achieved, with treatment maintenance ○ PASI 50 improvement—not achieved, treatment change regimen should be modified ○ PASI response is across 50 to 75%, therapy change(if DLQI>5), therapy maintenance (if DLQI ≤ 5).
French guideline of systemic therapy	Factors when establishing treatment goals for systemic therapy: ○ Disease severity ○ PsA/any comorbidity presence ○ Physical, social and psychological patient disease impact ○ The positive benefit-risk balance of ongoing systemic treatment ○ Oppinion and satisfaction level of patient	An adequate treatment response: PASI 75 (from the baseline), or PASI 50 with DLQI ≤ 5.
British guideline of systemic therapy	Treatment choice according to the patient and other factors: ○ Psoriasis features (therapeutic aim, disorder phenotype, activity pattern, impact and severity of disorder, PsA presence) ○ Other factors (age, weight, comorbidities—past/present, pregnancy, views/preferences on treatmentfrequency and administration way, adherence).	○ Assessment whether the minimal response have been met defined as: at least 50% decrease in baseline severity of disease (a response of PASI 50, otherwise percentage BSA) and improvement of physical, social or psychological performance (DLQI—4-point).
Spanish consensus of systemic therapy	The ideal outcome is to achieve: PASI 90 and a PGA ≤ 1, or as an alternative, a minimal, topical treatment controllable localized disease (PGA ≤ 2 and PASI < 5), DLQI ≤ 1, prolonged remissions without loss of efficacy, no worsening of comorbidities. Criteria for an appropriate response initially and in the long term (more than 6 months), 1 of: PASI 75, PASI < 5, PGA ≤ 1 and DLQI < 5. Criteria for the minimum efficacy required: PASI 50, PASI < 5.	Therapeutic failure during initiation of the treatment: ○ a score is equal to or greater than those constituting the criteria for moderate-to-severe psoriasis at the end of the induction phase ○ there is no adequate response according to the physician and the patient by the end of the induction phase, ○ a decrease in 50% from the baseline PASI score has not been achieved (or this degree of response has been lost) after the induction phase.

Abbreviations. PsA—psoriatic arthritis; PASI—Psoriasis Area and Severity Index; BSA—Body Surface Area; DLQI—dermatology life quality index; RCT—randomized controlled trial; PGA—Physician’s Global Assessment.

**Table 2 life-12-02050-t002:** Psoriasis frequency in difficult-to-treat areas with psoriasis severity [38].

Difficult-to-Treat Areas	% of Patients with Mild Psoriasis	% of Patients with Moderate Psoriasis	% of Patients with Severe Psoriasis
scalp	48.1%	57.8%	66.1%
face	27.6%	41.8%	53.3%
palms	13.5%	22.9%	19.5%
nails	25.6%	31.1%	42.4%
genitals	12.5%	18.1%	27.2%
soles	16.7%	24.7%	22.1%

**Table 3 life-12-02050-t003:** Examples of patients’ scores involve difficult-to-treat areas.

Main Area	Additional Areas	Severity Assessment
trunk BSA less than < 10% severity 4 for erythema, induration and scaling PASI = 3.6	scalp >70% severity 4 for erythema, induration and scaling	PASI = 8 → PSO moderate or PSSI = 48 → PSO severe
	nails matrix and nail bed completely affected 10 nails	PASI = 6 → PSO moderate or NAPSI = 80 → PSO severe
	palmo-plantar both hands or both soles severity 4 for erythema, induration, scaling, fissuring	PASI = 8 → PSO moderate or ESIF = 24 → PSO severe

Abbreviations: PASI—Psoriasis Area and Severity Index; BSA—Body Surface Area; PSSI—Psoriasis Scalp Severity Index; NAPSI—Nail Psoriasis Severity Index; ESIF—Erythema, Scaling, Induration, Fissuring.

**Table 4 life-12-02050-t004:** Treatment goals in psoriasis are defined by European guidelines.

	Severity Scores to Achieve	Quality of Life
Treatment goals (assessed after 10–16 weeks and then every 8 weeks treatment goals	PASI 90 or PASI ≤ 2 PGA clear or almost clear	DLQI < 2
minimum efficacy (lowest hurdle for treatment modification)	PASI 50	DLQI < 5 or DLQI improvement ≥ 5

Abbreviations: PASI—Psoriasis Area and Severity Index; PGA—Physician’s Global Assessment; DLQI—Dermatology Life Quality Index [47].

## Data Availability

Not applicable.

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
