# Peer review of "Psoriasis Management Challenges Regarding Difficult-to-Treat Areas: Therapeutic Decision and Effectiveness"

_life, 2022, doi:10.3390/life12122050_

Round 1
Reviewer 1 Report
In the present manuscript, authors have explored "Bias and challenges in the psoriasis management: implications of difficult-to-treat areas in therapeutic decision and evaluation of treatment effectiveness". The subject is of interest and falls in the topics of this journal “Life”. English can be improved. The manuscript is acceptable with minor revision.
After reviewing the manuscript thoroughly, I have following comments:
The full form of abbreviations is missing. Check these things through the manuscript and correct them.
The conclusion should be more informative. Improve it.
Author Response
Dear Reviewer 1,
I am writing to you in regards to the manuscript entitled “Bias and challenges in the psoriasis management: implications of difficult-to-treat areas in therapeutic decision and evaluation of treatment effectiveness”, which was submitted to your esteemed journal – Life, Special Issue: Current Research on Dermatology: Pathology, Clinical Manifestation, Investigation and Therapy, and was under review. This letter is a point-by-point response letter to the first reviewer’s comments. Please find below the issues and responses/corrections given.
Reviewer 1
Issue 1 - English can be improved.
Response: Thank you for your objective observation. Extensive language and grammar editing has been made throughout the manuscript so that it will meet the necessary language requirements.
Issue 2 – The full form of abbreviations is missing. Check these things through the manuscript and correct them.
Response: We are very appreciative for the extensive and detailed analysis of our work and have made the necessary changes to our manuscript by including all of the extended form of the abbreviations throughout the text.
Issue 3 – The conclusion should be more informative. Improve it.
Response: Thank you for the objective observation made; we have re-written the conclusions in the hopes of better suiting the reviewer’s excellence standards: “In clinical practice psoriasis severity is usually categorized as “mild”, “moderate”, and “severe”, using measurement tools which underestimate the actual disease severity in cases where skin lesions involve ‘special areas’ such as: the face, palms, soles, genitalia and scalp. It is necessary to consider the lesions’ location and patients’ quality of life in order to assess psoriasis severity more fully and accurately. The solutions considered for evaluating difficult-to-treat areas reside in establishing the severity according to the highest calculated score (PASI or PSSI, or NAPSI, or ESIF), and/or in the addition of a correction factor when calculating PASI for special areas. The algorithms used to monitor and evaluate therapeutic efficacy should be upgraded with emphasis on the diffi-cult-to-treat areas which influence greatly the overall severity degree.
The need to redefine disease severity with the inclusion of special areas is becoming increasingly obvious and should be done in a practical manner so that it is useful in both clinical and research settings.”
Response for Reviewer 1. Thank you for your suggestions. We added and included in the text all the necessary changes, corrections and information given.
Thank you for all your consideration!

Reviewer 2 Report
This paper is very interesting and well written. The authors underline the criticisms of the assessing the severity of psoriasis, especially when difficult to treat areas are involved. This carries the risk of underestimating disease severity and limited access to care for patients who do not meet current disease severity criteria. The proposals of integrating specific evaluation scales per site or of modifying the algorithms of the current assessments are well motivated and argued.
However I have some minor suggestions:
1. There are some typo errors.
· Please remove at the end of the lists in the text: many others or ect .. (lines 60, 96,442)
· Please do not start the sentence by writing a number but write the same number in words (lines 250,256)
2. Lines 227-228-229 are entered incorrectly in paragraph 3.7: The Psoriasis Scalp Severity Index (PSSI) and should be included in section 3.8: The Nail Psoriasis Severity Index (NAPSI)
3. For each rating scale, where it has not been carried out, add a brief explanation reporting the minimum and maximum values to provide essential information for the interpretation of the score.
4. I agree with the authors that one of the limitations of the PASI is to risk underestimating the severity of the disease when difficult to treat areas that are involved, because they are generally small areas and that weigh little on the total score, even if serious. It is also agreed that precisely for this limit it is difficult to use the PASI as a parameter to monitor the evolution of the disease in difficult to treat areas because the variations in the score can be slight. The third and fourth limitations of the PASI score are not well understood. Do the authors consider it necessary to distinguish the type of lesion (scaling, induration and redness), attributing a different degree of severity between these parameters regardless of their extent and localization? Can it be useful for determining the severity of the disease and / or for orienting towards targeted therapies? Excluding topical treatments, is there evidence on the effect of psoriasis treatments on different clinical manifestations in terms of specific improvement for scaling, induration and redness? Please clarify.
5. In line 250 it is not clear to what the percentages refer to. Please clarify
6. Paragraph 4: the authors stress the scarcity of studies on difficult areas. Provide a table summarizing RCTs, sub analyzes or real life trials for psoriasis in difficult-to-treat areas).
Author Response
Dear Reviewer 2,
I am writing to you in regards to the manuscript entitled “Bias and challenges in the psoriasis management: implications of difficult-to-treat areas in therapeutic decision and evaluation of treatment effectiveness”, which was submitted to your esteemed journal – Life, Special Issue: Current Research on Dermatology: Pathology, Clinical Manifestation, Investigation and Therapy, and was under review. This letter is a point-by-point response letter to the second reviewer’s comments. Please find below the issues and responses/corrections given.
Reviewer 2
Issue 1 – There are some typo errors. Please remove at the end of the lists in the text: many others or ect .. (lines 60, 96,442). Please do not start the sentence by writing a number but write the same number in words (lines 250,256).
Response: We are very appreciative of the in-depth analysis of our manuscript and have made the necessary changes to it in order to better suit the reviewer’s high standards of publishing. We have eliminated the double points from where indicated and where we have found them. At the same time, we have re-written the numbers at the beginning of the sentences with words, as indicated: “Sixty-four point 8 percent, 42.4% and 21.9% of patients had an involvement of more than one difficult-to-treat area. According to its severity (Table 2), the prevalence of psoriasis in the scalp, face, genitals, and nails was directly proportional to the severity of the psoriasis; for example, 66.1% of patients with severe psoriasis have lesions on the scalp. Among patients with mild psoriasis, 80.4% had an involvement of at least one difficult-to-treat area, whereas the prevalence among those with severe psoriasis increased to 89.0%. Sixty-eight point eight percent and 43.7% of patients suffering from severe psoriasis had at least 2 and, respectively, 3 difficult-to-treat areas. Outcome measures among patients with difficult-to-treat areas suggest that these patients could be defined as the population with the highest disease burden [38].”
Issue 2 – Lines 227-228-229 are entered incorrectly in paragraph 3.7: The Psoriasis Scalp Severity Index (PSSI) and should be included in section 3.8: The Nail Psoriasis Severity Index (NAPSI).
Response: Thank you for the objective observation made; we have made the necessary change: “3.6. The Psoriasis Scalp Severity Index (PSSI)
The PSSI measures the extent of psoriasis skin involvement and the severity of the erythema, infiltration, and desquamation of the scalp. The skin involvement and severity of psoriasis for the PSSI is scored by physicians using a scale ranging from 0 to 72, where 0 = no psoriasis, and higher scores indicating more severe disease involvement [32-39]. The PSSI calculation does not include the face or neck area. The scalp is a prevalent dif-ficult-to-treat area for psoriasis; the visibility of lesions and pruritus associated with scalp psoriasis may negatively affect the patients’ quality of life.
3.7. The Nail Psoriasis Severity Index (NAPSI)
The NAPSI is a numeric, reproducible, objective tool for the evaluation of nail pso-riasis, used to assess the severity of nail bed and matrix psoriasis by the area of in-volvement of the nail unit (each nail is divided into 4 quadrants and given a score for nail bed psoriasis and nail matrix psoriasis, depending on the presence of any of the features of nail psoriasis in that quadrant - pitting, leukonychia, red spots in the lunula, crumbling (0 for none, and 4 if present in 4 quadrants of the nail) and respectively, onycholysis, splinter hemorrhages, subungual hyperkeratosis, “oil drop” (0 for none, 4 for 4 quad-rants)). Each nail gets a matrix score and a nail bed score, the total of which represents the score for that nail (0-8); the sum of all the nails’ scores is the total NAPSI score (0-80 or 0-160 if the toenails are included in the calculation) [32].
Nail damage occurs in only 1 to 5% of patients, one in 2 patients with psoriasis being affected by nail psoriasis at any given time; an estimated lifetime incidence of the nail damage ranges between 80 to 90%.”
Issue 3 – For each rating scale, where it has not been carried out, add a brief explanation reporting the minimum and maximum values to provide essential information for the interpretation of the score.
Response: Thank you for the extensive and objective observations made in regards to our manuscript. We have made the necessary adjustments to the manuscript by including the information requested, as follows:
For Body surface area - BSA: “Psoriasis may be considered: mild if < 3% BSA, moderate 3% to 10% BSA, severe > 10% BSA [19].”
For Physician Global Assessment (PGA): “The PGA is another tool used in order to assess the severity of psoriasis by using , a 7-point scale, ranging from clear to severe.”
For the Psoriasis Area and Severity Index (PASI): “PASI <1 (minimal), PASI <3 (very mild), PASI <5 (mild), PASI 5-10 (moderate) and PASI >10 (severe)”
For the Investigator Global Assessment (IGA) scale: “The IGA scale is a visual assessment tool that consists of a score ranging from 0 (clear) to 4 (severe). The skin rated as having a score of 4 is bright red with marked plaque elevation and dominated by thick, non-tenacious scale.”
For the Dermatology Life Quality Index (DLQI): “The DLQI is a patient-reported questionnaire which was developed because the quality-of-life measures are helpful in demonstrating that changes in the severity of the skin lesions correspond to an improvement in the patients’ quality of life. The questions that make up the DLQI are scored on a four -point scale, as follows: 0, not at all/not rel-evant; 1, a little; 2, a lot; and 3, very much, giving a total DLQI score which can range from 0 to even 30. [26-28].”
For the Nail Psoriasis Severity Index (NAPSI): “The NAPSI is a numeric, reproducible, objective tool for the evaluation of nail pso-riasis, used to assess the severity of nail bed and matrix psoriasis by the area of in-volvement of the nail unit (each nail is divided into 4 quadrants and given a score for nail bed psoriasis and nail matrix psoriasis, depending on the presence of any of the features of nail psoriasis in that quadrant - pitting, leukonychia, red spots in the lunula, crumbling (0 for none, and 4 if present in 4 quadrants of the nail) and respectively, onycholysis, splinter hemorrhages, subungual hyperkeratosis, “oil drop” (0 for none, 4 for 4 quad-rants)). Each nail gets a matrix score and a nail bed score, the total of which represents the score for that nail (0-8); the sum of all the nails’ scores is the total NAPSI score (0-80 or 0-160 if the toenails are included in the calculation) [32].”
For the Erythema, Scaling, Induration, and Fissuring (ESIF): “ESIF is assessed for palm and sole psoriasis using a 4-point scale (from 0 = clear to 3 = severe) and is calculated by adding the scores for the 4 signs for the soles of the feet, yielding a total range from 0 (no disease) to 24 (most severe condition) [37].”
Issue 4 - I agree with the authors that one of the limitations of the PASI is to risk underestimating the severity of the disease when difficult to treat areas that are involved, because they are generally small areas and that weigh little on the total score, even if serious. It is also agreed that precisely for this limit it is difficult to use the PASI as a parameter to monitor the evolution of the disease in difficult to treat areas because the variations in the score can be slight. The third and fourth limitations of the PASI score are not well understood. Do the authors consider it necessary to distinguish the type of lesion (scaling, induration and redness), attributing a different degree of severity between these parameters regardless of their extent and localization? Can it be useful for determining the severity of the disease and / or for orienting towards targeted therapies? Excluding topical treatments, is there evidence on the effect of psoriasis treatments on different clinical manifestations in terms of specific improvement for scaling, induration and redness? Please clarify.
Response: Thank you for the deep analysis and understanding of our manuscript. We wish to highlight the fact that the PASI score can be improved in its use of psoriasis lesions evaluation by shedding light on its limitations; the first two ones being: “The first limitation comes from the specific PASI formula and location of lesions for hard-to-treat psoriasis. Considering the case of a patient with 3 affected areas, with minimal trunk involvement and a moderate-severe involvement of the scalp and/or face – the severity of the score using PASI could be deficient and lead to an under-treatment of the patient. The second limitation – the estimation of the percentage of body surface area (BSA) involvement, in which case there is high variability particularly in those patients suffering from limited psoriasis lesions, resulting in a poor detection of changes in mild or moderate psoriasis.” The third limitation “refers to patients with very different clinical manifestations of psoriasis, having the same PASI score; for example: one has widespread but mild severity psoriasis lesions and another has localized but severe psoriasis lesions” – we believe that the score can be influenced easily by the different parameter degrees/intensity (scaling, induration and redness) with important impact from the extent of the lesion; a lesion which is localized but severe in parameters might have equal PASI score as lesions that are milder, but extensive. The fourth limitation refers to: “Because plaque elevation, scaling and erythema are rated equally, treatments that temporarily affect scaling or erythema will affect the PASI score more than equally effective treatments that do not – the latter being the fourth limitation.” – treatments which improve one of the parameters from the PASI score can modify the total score in a lesser or more important extent, as compared to treatments which influence other parameters. Different treatments might have different effects on some or all of the parameters of the PASI score, influencing the total sum and giving confounding results. It could be of great use the possibility of targeting the lesional aspects/parameters according to each of the parameter’s severity. As to what addresses the therapeutic effect in regards to each PASI parameter, there is no study that addresses this matter that we are knowledgeable of.
Issue 5 - In line 250 it is not clear to what the percentages refer to. Please clarify
Response: We are appreciative of the observation made and have clarified the issue, as it is revealed in the following paragraph taken from the manuscript: “A cross-sectional study that included more than 4000 adults with psoriasis from the Danish Skin Cohort, evaluated the involvement of hard-to-treat areas. The most frequently difficult-to-treat areas are the scalp (in 43.0% of patients), the face (29.9%), nails (24.5%), soles (15.6%), genitals (14.1%), and palms (13.7%). Sixty-four point 8 percent, 42.4% and 21.9% of patients had an involvement of at least one, and respectively at least 2 and at least three difficult-to-treat areas. According to its severity (Table 2), the prevalence of psoriasis in the scalp, face, genitals, and nails was directly proportional to the severity of the psoriasis; for example, 66.1% of patients with severe psoriasis have lesions on the scalp. Among patients with mild psoriasis, 80.4% had an involvement of at least one difficult-to-treat area, whereas the prevalence among those with severe psoriasis increased to 89.0%. Sixty-eight point eight percent and 43.7% of patients suffering from severe psoriasis had at least 2 and, respectively, 3 difficult-to-treat areas. Outcome measures among patients with difficult-to-treat areas suggest that these patients could be defined as the population with the highest disease burden [38].”
Issue 6 - Paragraph 4: the authors stress the scarcity of studies on difficult areas. Provide a table summarizing RCTs, sub analyzes or real life trials for psoriasis in difficult-to-treat areas).
Response: Thank you for the analysis provided for our manuscript and are deeply grateful for the subjects raised, giving us the possibility to better our results. As such, we would like to further highlight the fact that our manuscript is supported by studies included and debated in the results section, throughout it, by references 38, 39, 44 to 47 and 53 to 57, as highlighted below, revealing results from studies carried out in 2002, 2015, 2017, 2018, 2019, 2020, 2021; our manuscript would suffer from redundancy and monotonous information if it should be repeated, making it tedious to read:
- Egeberg, A.; See, K.; Garrelts, A.; Burge, R. Epidemiology of psoriasis in hard-to-treat body locations: data from the Danish skin cohort. BMC Dermatology 2020, 20, 3.
- Callis Duffin, K.; Mason, M.A.; Gordon, K.; Harrison, R.W.; Crabtree, M.M.; Guana, A.; Germino, R.; Lebwohl, M. Char-acterization of Patients with Psoriasis in Challenging-to-Treat Body Areas in the Corrona Psoriasis Registry. Dermatology 2021, 237(1), 46-55..
- Mosca, M.; Hong, J.; Hadeler, E.; Brownstone, N.; Bhutani, T.; Liao, W. Scalp Psoriasis: A Literature Review of Effective Therapies and Updated Recommendations for Practical Management. Dermatol Ther (Heidelb)2021, 11, 769–797.
- Arif, A.; Mahadi, I.D.R.; Yosi, A. Correlation between nail psoriasis severity index score with quality of life in nail psoriasis. Bali MedJ 2021, 10(1), 256-260.
- Timotijević, Z.S.; Trajković, G.; Jankovic, J.; Relić, M.; Đorić, D.; Vukićević, D.; Relić, G.; Rašić, D.; Filipović, M.; Janković, S. How frequently does palmoplantar psoriasis affect the palms and/or soles? A systematic review and meta-analysis. Adv Dermatol Allergol 2019, 36(5), 595–603.
- Kumar, B.; Saraswat, A.; Kaur, I. Palmoplantar lesions in psoriasis: a study of 3065 patients. Acta Derm Venereol 2002, 82, 192-5.
- Gordon, K.B.; Strober, B.; Lebwohl, M.; Augustin, M.; Blauvelt, A.; Poulin, Y.; Papp, K.A.; Sofen, H.; Puig, L.; Foley, P.; Ohtsuki, M.; Flack, M.; Geng, Z.; Gu, Y.; Valdes, J.M.; Thompson, E.H.Z.; Bachelez, H. Efficacy and safety of risankizumab in moderate-to-severe plaque psoriasis (UltIMMa-1 and UltIMMa-2): results from two double-blind, randomised, place-bo-controlled and ustekinumab-controlled phase 3 trials. Lancet 2018, 392, 651-61.
- Papp, K.; Thaçi, D.; Reich, K.; Riedl, E; Langley, R.G.; Krueger, J.G.; Gottlieb, A.B.; Nakagawa, H.; Bowman, E.P.; Mehta, A.; Li, Q.; Zhou, Y.; Shames, R. Tildrakizumab (MK3222), an anti-interleukin-23p19 monoclonal antibody, improves psoriasis in a phase IIb randomized placebo-controlled trial. Br J Dermatol. 2015, 173(4), 930–9.
- Blauvelt, A.; Papp, K.A.; Griffiths, C.E.; Randazzo, B.; Wasfi, Y.; Shen, Y.K.; Li, S.; Kimball, A.B. Efficacy and safety of guselkumab, an anti-interleukin-23 monoclonal antibody, compared with adalimumab for the continuous treatment of patients with moderate to severe psoriasis: results from the phase III, double-blinded, placebo- and active comparator controlled VOYAGE 1 trial. J Am Acad Dermatol. 2017, 76(3), 405–17.
- Elewski, B.E.; Puig, L.; Mordin, M.; Gilloteau, I.; Sherif, B.; Fox, T.; Gnanasakthy, A.; Papavassilis, C.; Strober, B.E. Psoriasis patients with psoriasis Area and Severity Index (PASI) 90 response achieve greater health-related quality-of-life im-provements than those with PASI 75-89 response: results from two phase 3 studies of secukinumab. J Dermatolog Treat. 2017, 28(6), 492–99.
- Carretero, G.; Puig, L.; Carrascosa, J.M.; Ferrándiz, L.; Ruiz-Villaverde, R.; de la Cueva, P.; Belinchon, I.; Vilarrasa, E.; Del Rio, R.; Sánchez-Carazo, J.L.; López-Ferrer, A.; Peral, F.; Armesto, S.; Eiris, N.; Mitxelena, J.; Vilar-Alejo, J.; A Martin, M.; Soria, C.; from the Spanish Group of Psoriasis. Redefining the therapeutic objective in psoriatic patients’ candidates for biological therapy. J Dermatolog Treat. 2018, 29(4), 334–46.
Response for Reviewer 2: Thank you for your suggestions. We added and included in the text all your suggested changes and necessary information.
Thank you for all your consideration!

Reviewer 3 Report
The article has sufficient structure. However, a thorough revision of the language is required, some passages are not completely understandable.
The abstract is long, it should be summarized further.
The introduction should be considerably expanded, it appears particularly short.
Materials and methods raises some concerns. Whatever the type of review, it would be more correct to specify in detail the entire review process from the definition of the research questions to the selection of the papers to be included. It would be necessary to describe the criteria for inclusion and exclusion of the papers.
The results are well presented and analyzed.
The discussion needs to be broadened considerably, without a good comparison with other studies, a paper does not have great relevance.
A review, then, requires the presence of a paragraph on the advantages and limitations of the study.
The conclusions are generic, it is necessary to focus them more.
Author Response
Dear Reviewer 3,
I am writing to you in regards to the manuscript entitled “Bias and challenges in the psoriasis management: implications of difficult-to-treat areas in therapeutic decision and evaluation of treatment effectiveness”, which was submitted to your esteemed journal – Life, Special Issue: Current Research on Dermatology: Pathology, Clinical Manifestation, Investigation and Therapy, and was under review. This letter is a point-by-point response letter to the third reviewer’s comments. Please find below the issues and responses/corrections given.
Reviewer 3
Issue 1 – The article has sufficient structure. However, a thorough revision of the language is required, some passages are not completely understandable.
Response: Thank you for your objective observation. Extensive language and grammar editing has been made throughout the manuscript so that it will meet the necessary language requirements.
Issue 2 – The abstract is long, it should be summarized further.
Response: We are thankful for the observation and the detailed analysis of our manuscript. As such, we have adapted the abstract so that it better suits the reviewer’s standards of reporting and the manuscript’s main body of text, as follows: “Psoriasis is not optimally controlled in spite of newly developed treatments, possibly due to the difficulty of objectively quantifying the disease's severity, considering the limitations of the clinical scores used in clinical practice. A major challenge addresses difficult-to-treat areas, especially in the absence of significant body surface involvement. It is controversial whether the severity evaluation of patients with several affected areas (having at least one difficult-to-treat area) should be done differently from current methods. Scores used for special areas (PSSI, NAPSI and ESIF) allow an accurate assessment of disease severity in difficult-to-treat areas, but the issue of whether to integrate these scores into PASI, BSA or DLQI, remains. The aim of our review is to provide an overview of the main current issues in determining psoriasis severity in patients with psoriasis in difficult-to-treat areas, and suggest possible solutions for the optimal integration of the area assessment in current scores: severity can be either established according to the highest cal-culated score (PASI or PSSI or NAPSI or ESIF) or by adding a correction factor in the calculation of PASI for special areas.”
Issue 3 – The introduction should be considerably expanded, it appears particularly short.
Response: We appreciate the objective observation and would like to highlight the extended information introduced in the specific part of our manuscript, with relevant references for further support, as follows: “Psoriasis is a chronic systemic inflammatory disease which affects the skin, 30% of cases having joint involvement (psoriatic arthritis) [1,2]. Psoriasis management brings many challenges including an increased prevalence over the past years, chronicity, high grade of disability for some cases and associated comorbidities [3]. Psoriasis plaques are more common on the elbows and knees but can affect any area of the body including the scalp, groins, and genital area. Nails are frequently affected and can be seen as an isolated location [1,2]. Genetic factors are various [1-3], and other factors that may trigger or ex-acerbate psoriasis can include: stress, body mass index (BMI), infection, drugs (be-ta-blockers, lithium, angiotensin-converting enzyme (ACE)-inhibitors, (synthetic) anti-malarials, tetracyclines, non-steroidal anti-inflammatory drugs (NSAIDs)), withdrawal of systemic (or even potent local) corticosteroids, chronic alcohol consumption, smoking, friction, minimal trauma of the skin – even a slight irritation, radiotherapy, endocrine disorders, and many other [3-6].
Psoriasis represents a constant public health challenge, affecting approximately 125 million people globally [7], although 81% of the world’s countries are lacking in psoriasis epidemiology [8]. The prevalence within adult populations estimates in between 0.91% in the U.S.A. and 8.5% in Norway, increasing globally from 758 cases per 100,000 in 1990 to 812 per 100,000 in 2017. The highest incidence rates are in the European region, where an increase in incidence from 143.7 cases per 100,000 in 1990 to 147.2 in 2017 was reported [4,9,10]. This supports the statement that there is great variation in the prevalence rates among the world’s regions, with variations from 0,73% (to even 2,9%) in Europe, or 0,7 (even 2,6%) in the United States, to under 0,5% in Asia (China, Sri Lanka), Latin America, India, or Africa (Tanzania, Egypt) [11].
In Romania, psoriasis is affecting approximately 400,000 people, as evidenced by the first epidemiological study published recently in 2021 [12]. The prevalence of psoriasis vulgaris within this study was 4.99% [12] (although there are other sources reporting percentages as high as 5,18) [13]; this study is currently continuing in order to assess factors contributing to the increase in psoriasis prevalence over time [12].
Psoriasis displays heterogeneity in clinical presentation and treatment response, varying with disease duration, the areas of involvement and percentage of body surface affected. Perception of psoriasis severity differs at baseline from long-term disease course. Disease severity guides treatment decisions, choice of medication and the intensity of the treatment response, as well as eligibility criteria for participation in phase II or III clinical studies. In clinical practice, psoriasis severity assessment usually combines objective and subjective parameters, including the extent of skin involvement, signs and symptoms, and also the specific location of lesions and the impact on every patients’ quality of life [14,15].
There is great variability in quantifying the severity of the disease worldwide and we aimed at evaluating the current guidelines/recommendations and the severity scores in order to achieve better results as an integrated approach to the disease by combining the existing scores in current therapeutic guidelines, recommendations and protocols.”.
Issue 4 – Materials and methods raises some concerns. Whatever the type of review, it would be more correct to specify in detail the entire review process from the definition of the research questions to the selection of the papers to be included. It would be necessary to describe the criteria for inclusion and exclusion of the papers.
Response: This manuscript is the result of our need to bring together base-information and updated materials on psoriasis and its difficult-to-treat areas. We would like to further include the updated Materials and methods section, hopefully meeting with the reviewer’s analysis and concerns: “The current work is the result of conducting a thorough comprehensive analysis of the current specialty literature, generating a narrative review type of article by using updated materials starting from the year 2000 and beyond, with exclusion of prior materials and data reported in languages other than English. Database searches were done using Pubmed, Google Scholar and Web of Science, using keywords such as: “psoriasis”, “psoriasis scores”, “PASI”, “BSA”, “PGA”, “IGA”, “DLQI”, “PSSI”, “NAPSI”, “ESIF” and “difficult-to-treat areas”, individually or combined, generating the current work after selection of the most representative, relevant and work-related, pertinent articles. After searching three databases and zero registers, eliminating articles irrelevant for the current subject, based on 47 relevant results, the following review was compiled.”
Issue 5 – The discussion needs to be broadened considerably, without a good comparison with other studies, a paper does not have great relevance. A review, then, requires the presence of a paragraph on the advantages and limitations of the study.
Response: Thank you for the objective observations made throughout the review. We further include the updated material in our discussion section, with advantages and limitations to our review, along with a figure which summarizes the given information, highlighting the most important information from our manuscript: ” Psoriasis of the scalp, face, intertriginous areas, genitals, hands, feet, and nails is of-ten underdiagnosed, and under-treated. In spite of the small surface area which is com-monly affected by psoriatic lesions in these locations, patients have disproportionate levels of physical impairment and emotional distress. Limitations in current disease se-verity scores do not fully assess the impact of disease on a patient’s quality of life, many patients not receiving adequate care. In these cases, the therapeutic attitude to adapt is to adapt therapy (dose increase, therapeutic combination), change of treatment (switch) or continuation of treatment for the next 3 months with reevaluation [59]. Psoriasis severity classification is a major problem which needs to be addressed in order to guide the phy-sician’s decisions regarding treatment, having always in mind that the disease is hetero-geneous in clinical expression and response to treatment, in its duration and involved areas (including percentage of body area), being constantly variable [22].
Psoriasis is highly influenced by external factors; the list of triggering factors for flare-ups is extensive, starting with stress, mild localized skin trauma, different infections and drugs for different co-morbidities, alcohol consumption, smoking, weather etc. Seventy three percent of the patients have at least one comorbidity that can influence the disease evolution and response to treatment, especially for difficult-to-treat areas [59]. In the given example – the patient with 2 areas affected by psoriasis has been evaluated after 3 months of treatment with ∆ PASI 90 for lesions on the trunk and ∆ PASI < 50 for scalp. The patient suffered a great level of stress during these 3 months but according to DLQI there was significant improvement.
There are many systems classifying psoriasis severity, but none have managed to reach a consensus, not having clear-cut demarcation between severity degrees (moderate and severe), the methodology disregarding the involvement of psoriasis in difficult-to-treat areas; the fact that current practices employ systems that are mainly used by physicians/dermatologists (such as PASI, PGA and/or BSI) and by patients (DLQI) alike is a step further in psoriasis severity assessment and treatment, the quality of life being of paramount importance. [22]. Figure 2 reveals the raised issues, with point-by-point ex-planations and proposed solutions for the clinicians in their daily practice.
Figure 2. Issues and possible solutions in psoriasis severity effective calculation.
Abbreviations: PASI - Psoriasis Area and Severity Index; BSA - Body Surface Area; PSSI - Psoriasis Scalp Severity Index; NAPSI - Nail Psoriasis Severity Index; ESIF - Erythema, Scaling, Induration, Fissuring.
For psoriasis involving difficult-to-treat areas the challenge extends from establishing severity and choosing the optimal treatment to correctly evaluating the efficacy or failure of the therapy; this is a major problem which our paper addressed, having the advantage of bringing forth a practical, clinical issue which dermatologists face. There are several solutions to this issue; the first one is the evaluation of different scores for each area and the success or failure of the treatment to be evaluated by the area with the smallest improvement. The second solution is the inclusion of special areas in the treatment goal algorithm or to use a completely different algorithm for psoriasis in difficult-to-treat areas by using the data obtained in the different clinical studies or real-world experience. The current paper has made a summary of the literature at hand, examining the current knowledge on psoriasis and its score assessment, with the limitation of possibly having a broad area of research data (a wide range of psoriasis score reporting analysis), but with the advantage of identifying possible new research areas.”
We would like to further highlight the fact that our manuscript is supported by studies included in the results section, throughout it, by references 38, 39, 44 to 47 and 53 to 57, as highlighted below, revealing results from studies carried out in 2002, 2015, 2017, 2018, 2019, 2020, 2021:
- Egeberg, A.; See, K.; Garrelts, A.; Burge, R. Epidemiology of psoriasis in hard-to-treat body locations: data from the Danish skin cohort. BMC Dermatology 2020, 20, 3.
- Callis Duffin, K.; Mason, M.A.; Gordon, K.; Harrison, R.W.; Crabtree, M.M.; Guana, A.; Germino, R.; Lebwohl, M. Char-acterization of Patients with Psoriasis in Challenging-to-Treat Body Areas in the Corrona Psoriasis Registry. Dermatology 2021, 237(1), 46-55..
- Mosca, M.; Hong, J.; Hadeler, E.; Brownstone, N.; Bhutani, T.; Liao, W. Scalp Psoriasis: A Literature Review of Effective Therapies and Updated Recommendations for Practical Management. Dermatol Ther (Heidelb)2021, 11, 769–797.
- Arif, A.; Mahadi, I.D.R.; Yosi, A. Correlation between nail psoriasis severity index score with quality of life in nail psoriasis. Bali MedJ 2021, 10(1), 256-260.
- Timotijević, Z.S.; Trajković, G.; Jankovic, J.; Relić, M.; Đorić, D.; Vukićević, D.; Relić, G.; Rašić, D.; Filipović, M.; Janković, S. How frequently does palmoplantar psoriasis affect the palms and/or soles? A systematic review and meta-analysis. Adv Dermatol Allergol 2019, 36(5), 595–603.
- Kumar, B.; Saraswat, A.; Kaur, I. Palmoplantar lesions in psoriasis: a study of 3065 patients. Acta Derm Venereol 2002, 82, 192-5.
- Gordon, K.B.; Strober, B.; Lebwohl, M.; Augustin, M.; Blauvelt, A.; Poulin, Y.; Papp, K.A.; Sofen, H.; Puig, L.; Foley, P.; Ohtsuki, M.; Flack, M.; Geng, Z.; Gu, Y.; Valdes, J.M.; Thompson, E.H.Z.; Bachelez, H. Efficacy and safety of risankizumab in moderate-to-severe plaque psoriasis (UltIMMa-1 and UltIMMa-2): results from two double-blind, randomised, place-bo-controlled and ustekinumab-controlled phase 3 trials. Lancet 2018, 392, 651-61.
- Papp, K.; Thaçi, D.; Reich, K.; Riedl, E; Langley, R.G.; Krueger, J.G.; Gottlieb, A.B.; Nakagawa, H.; Bowman, E.P.; Mehta, A.; Li, Q.; Zhou, Y.; Shames, R. Tildrakizumab (MK3222), an anti-interleukin-23p19 monoclonal antibody, improves psoriasis in a phase IIb randomized placebo-controlled trial. Br J Dermatol. 2015, 173(4), 930–9.
- Blauvelt, A.; Papp, K.A.; Griffiths, C.E.; Randazzo, B.; Wasfi, Y.; Shen, Y.K.; Li, S.; Kimball, A.B. Efficacy and safety of guselkumab, an anti-interleukin-23 monoclonal antibody, compared with adalimumab for the continuous treatment of patients with moderate to severe psoriasis: results from the phase III, double-blinded, placebo- and active comparator controlled VOYAGE 1 trial. J Am Acad Dermatol. 2017, 76(3), 405–17.
- Elewski, B.E.; Puig, L.; Mordin, M.; Gilloteau, I.; Sherif, B.; Fox, T.; Gnanasakthy, A.; Papavassilis, C.; Strober, B.E. Psoriasis patients with psoriasis Area and Severity Index (PASI) 90 response achieve greater health-related quality-of-life im-provements than those with PASI 75-89 response: results from two phase 3 studies of secukinumab. J Dermatolog Treat. 2017, 28(6), 492–99.
- Carretero, G.; Puig, L.; Carrascosa, J.M.; Ferrándiz, L.; Ruiz-Villaverde, R.; de la Cueva, P.; Belinchon, I.; Vilarrasa, E.; Del Rio, R.; Sánchez-Carazo, J.L.; López-Ferrer, A.; Peral, F.; Armesto, S.; Eiris, N.; Mitxelena, J.; Vilar-Alejo, J.; A Martin, M.; Soria, C.; from the Spanish Group of Psoriasis. Redefining the therapeutic objective in psoriatic patients’ candidates for biological therapy. J Dermatolog Treat. 2018, 29(4), 334–46.
Issue 8 – The conclusions are generic, it is necessary to focus them more.
Response: We very much appreciate the observation and would like to highlight the fact that we have changed the conclusions part of our manuscript in the hopes of meeting with the reviewer’s excellence standards: “In clinical practice psoriasis severity is usually categorized as “mild”, “moderate”, and “severe”, using measurement tools which underestimate the actual disease severity in cases where skin lesions involve ‘special areas’ such as: the face, palms, soles, genitalia and scalp. It is necessary to consider the lesions’ location and patients’ quality of life in order to assess psoriasis severity more fully and accurately. The solutions considered for evaluating difficult-to-treat areas reside in establishing the severity according to the highest calculated score (PASI or PSSI, or NAPSI, or ESIF), and/or in the addition of a correction factor when calculating PASI for special areas. The algorithms used to monitor and evaluate therapeutic efficacy should be upgraded with emphasis on the diffi-cult-to-treat areas which influence greatly the overall severity degree.
The need to redefine disease severity with the inclusion of special areas is becoming increasingly obvious and should be done in a practical manner so that it is useful in both clinical and research settings.”
Response for Reviewer 3. Thank you for your suggestions. We added and included in the text all the necessary changes, corrections and information given.
Thank you for all your consideration!

Reviewer 4 Report
Dear Authors,
I read with interest your article. It is a good work on an interesting topic. However, some points must be improved for publication.
- The article is too long, with some important parts written too briefly, and other parts with too much text and being too redundant. This point is crucial to make the article proper for publication. This is emphasized below
- I would suggest to simplify the title, so as to make it more clearly understandable.
- Methods section: This is the main point to improve. This section is too short and lacks important information. It should include clearly the years included in the review, avoiding the use of "such as" which is not scientific. The specific words used in the search or the search command should be clearly stated. The way the articles were included should be clearly stated (if they were included after reading the title, both the title and the abstract, if all types of epidemiological studies were included, if studies were discarded because of language...).
- Lines 91-100; 101-106: reference needed.
- Line 126: It is wrongly written because it is a question.
- Lines 227-229: I think this paragraph should be included in the next section.
- Results section: This is the another point to improve. This section is too long and it is difficult to read it in a friendly way. Reducing the length or the article would be of great importance. For example, the section on DLQI is full of personal interpretations and redundant data. As well as the text between lines 243 and 330, which should be forcefully reduced.
- Adding one table or figure summarizing the possible solutions the author propose to the problems analyzed in the review would be of great interest.
Author Response
Dear Reviewer 4,
I am writing to you in regards to the manuscript entitled “Bias and challenges in the psoriasis management: implications of difficult-to-treat areas in therapeutic decision and evaluation of treatment effectiveness”, which was submitted to your esteemed journal – Life, Special Issue: Current Research on Dermatology: Pathology, Clinical Manifestation, Investigation and Therapy, and was under review. This letter is a point-by-point response letter to the fourth reviewer’s comments. Please find below the issues and responses/corrections given.
Reviewer 4
Issue 1 – The article is too long, with some important parts written too briefly, and other parts with too much text and being too redundant. This point is crucial to make the article proper for publication. This is emphasized below. I would suggest to simplify the title, so as to make it more clearly understandable.
Response: We appreciate the very objective observation and have shortened our manuscript (as it will be emphasized in the following issues to which we have responded point-by-point) and have also shortened the title of our manuscript: “Psoriasis management challenges regarding difficult-to-treat areas: therapeutic decision and effectiveness”.
Issue 2 – Methods section: This is the main point to improve. This section is too short and lacks important information. It should include clearly the years included in the review, avoiding the use of "such as" which is not scientific. The specific words used in the search or the search command should be clearly stated. The way the articles were included should be clearly stated (if they were included after reading the title, both the title and the abstract, if all types of epidemiological studies were included, if studies were discarded because of language...).
Response: This manuscript is the result of our need to bring together base-information and updated materials on psoriasis and its difficult-to-treat areas. We would like to further include the updated Materials and methods section, hopefully meeting with the reviewer’s analysis and concerns: “The current work is the result of conducting a thorough comprehensive analysis of the current specialty literature, generating a narrative review type of article by using updated materials starting from the year 2000 and beyond, with exclusion of prior materials and data reported in languages other than English. Database searches were done using Pubmed, Google Scholar and Web of Science, using keywords such as: “psoriasis”, “psoriasis scores”, “PASI”, “BSA”, “PGA”, “IGA”, “DLQI”, “PSSI”, “NAPSI”, “ESIF” and “difficult-to-treat areas”, individually or combined, generating the current work after selection of the most representative, relevant and work-related, pertinent articles. After searching three databases and zero registers, eliminating articles irrelevant for the current subject, based on 47 relevant results, the following review was compiled.”
Issue 3 – Lines 91-100; 101-106: reference needed.
Response: Thank you for the objective observation which we very much appreciate. We have adapted this information with references added to better suit our manuscript “In the past decade, new treatments and treatment strategies became available, especially for patients with moderate-to-severe disease. Among these newly developed treatments, biologics provide targeted inhibition of immune-mediated pathways in-volving cytokines, such as tumor necrosis factor (TNF), interleukin 17 (IL-17), IL-23, IL-36, etc. [16,17]. Besides all of these advancements, psoriasis is still not always optimally treated; the patient satisfaction rate with the existing therapies remains modest, all the while the disease burden being at high levels, in spite of the effectiveness of new treatments [17,18].”
Issue 4 – Line 126: It is wrongly written because it is a question.
Response: We are very appreciative of the careful consideration and analysis of our manuscript, and have made the necessary changes as follows: “When using the percentage of BSA as an indicator for psoriasis severity, it is im-portant that the measurement be made as accurate as possible.”
Issue 5 – Lines 227-229: I think this paragraph should be included in the next section.
Response: Thank you for the objective and oriented analysis given. We have made the change in our manuscript, as follows: “4. The challenges of individualized treatment and evaluating the treatment success
The issue of psoriasis in “difficult-to-treat” areas - Psoriasis localized on special ar-eas of the body is challenging to treat and is often associated with substantial physical discomfort and disability. “Difficult-to-treat” areas is a term which has been used by several authors in describing the psoriasis on the scalp, palms and soles, and nail, being usually associated with an increased emotional and functional impact [20-25]. Some au-thors also include as part of this classification the psoriasis of the face and inverse psori-asis [25].”
Issue 6 – Results section: This is the another point to improve. This section is too long and it is difficult to read it in a friendly way. Reducing the length or the article would be of great importance. For example, the section on DLQI is full of personal interpretations and redundant data. As well as the text between lines 243 and 330, which should be forcefully reduced.
Response: We are grateful for such in-depth analysis of the current paper and would like to introduce further the passages in our manuscript which were altered in order to meet with the necessary requirements:
“3.7. The Nail Psoriasis Severity Index (NAPSI)
The NAPSI is a numeric, reproducible, objective tool for the evaluation of nail pso-riasis, used to assess the severity of nail bed and matrix psoriasis by the area of in-volvement of the nail unit (each nail is divided into 4 quadrants and given a score for nail bed psoriasis and nail matrix psoriasis, depending on the presence of any of the features of nail psoriasis in that quadrant - pitting, leukonychia, red spots in the lunula, crumbling (0 for none, and 4 if present in 4 quadrants of the nail) and respectively, onycholysis, splinter hemorrhages, subungual hyperkeratosis, “oil drop” (0 for none, 4 for 4 quad-rants)). Each nail gets a matrix score and a nail bed score, the total of which represents the score for that nail (0-8); the sum of all the nails’ scores is the total NAPSI score (0-80 or 0-160 if the toenails are included in the calculation) [20].
3.8. The Erythema, Scaling, Induration, and Fissuring (ESIF)
ESIF is assessed for palm and sole psoriasis using a 4-point scale (from 0 = clear to 3 = severe) and is calculated by adding the scores for the 4 signs for the soles of the feet, yielding a total range from 0 (no disease) to 24 (most severe condition) [25].
A cross-sectional study that included more than 4000 adults with psoriasis from the Danish Skin Cohort, evaluated the involvement of hard-to-treat areas. The most frequently difficult-to-treat areas are the scalp (43.0%), the face (29.9%), nails (24.5%), soles (15.6%), genitals (14.1%), and palms (13.7%). 64.8%, 42.4% and 21.9% of patients had an involvement of more than one difficult-to-treat area. According to its severity (Table 2), the prevalence of psoriasis in the scalp, face, genitals, and nails was directly proportional to the severity of the psoriasis; for example, 66.1% of patients with severe psoriasis have lesions on the scalp. Among patients with mild psoriasis, 80.4% had an involvement of at least one difficult-to-treat area, whereas the prevalence among those with severe psoriasis increased to 89.0%. 68.8% and 43.7% of patients suffering from severe psoriasis had at least 2 and, respectively, 3 difficult-to-treat areas. Outcome measures among patients with difficult-to-treat areas suggest that these patients could be defined as the population with the highest disease burden [26].
Table 2. Prevalence of psoriasis in difficult-to-treat areas across psoriasis severity (26).
difficult-to-treat areas |
% of patients with mild psoriasis |
% of patients with moderate psoriasis |
% of patients with severe psoriasis |
scalp |
48.1% |
57.8% |
66.1% |
face |
27.6% |
41.8% |
53.3% |
palms |
13.5% |
22.9% |
19.5% |
nails |
25.6% |
31.1% |
42.4% |
genitals |
12.5% |
18.1% |
27.2% |
soles |
16.7% |
24.7% |
22.1% |
These difficult-to-treat areas have limited degree response to topical treatment and could be classified as moderate to severe psoriasis, even if BSA ≤10 and PASI ≤10. Data collected from the Corrona Psoriasis Registry revealed that 2/3 of psoriasis patients un-dergoing biological treatment have psoriatic arthritis and/or at least one form of psoriasis in an area difficult to treat (scalp, nail psoriasis, palmo-plantar). Scores dedicated to these special areas (PSSI, NAPSI, ESIF) allow a precise calculation of disease severity, but they are not integrated in the more commonly used scores like BSA, PGA, PASI [27].
The biggest challenge in establishing the disease severity, therapeutic choice and ef-ficiency monitoring are “difficult-to-treat” areas, especially in the absence of significant involvement of the body surface elsewhere. The location and morphological features of scalp, nail, palmo-plantar and genital psoriasis can often lead to ineffective topical treatment and often requires systemic treatment [28-31].
Determining psoriasis severity in patients with psoriasis in difficult-to-treat areas can be a demanding task because some definitions of severity depend on the percentage of surface area involved. Dedicated scores allow a more accurate calculation of disease severity in special areas, but the dilemma remains whether or not to assess the overall severity of the disease by integrating these special area scores into scores like PASI, PGA, BSA and DLQI. It is debatable whether in patients with several affected areas (a common situation in clinical practice, with at least one special area) the classification of the degree of severity should be made differently from current methods, or not.
We have an example of a psoriasis patient having 2 involved skin areas: scalp and trunk (Table 3). The scalp has erythema, induration and exfoliation of over 70%. Trunk lesions extend up to 10% with a severity of 4. The PASI score for this patient is 8, indi-cating a moderate form of disease; if in this patient’s case we would use the PSSI score (Psoriasis Scalp Severity Index), the value will add up to 60 points, indicating a severe form according to the European guidelines. Severity framing can significantly influence therapeutic options and the choice of effective treatment for both areas involved [32].
Table 3. Examples of patients’ scores involving difficult-to-treat areas.
Main area |
Additional areas |
Severity assessment |
trunk
BSA less than < 10%
severity 4 for erythema,
induration and scaling
PASI = 3.6 |
scalp >70% severity 4 for erythema, induration and scaling |
PASI = 8 → PSO moderate or PSSI = 48 → PSO severe |
nails matrix and nail bed completely affected 10 nails |
PASI = 6 → PSO moderate or NAPSI = 80 → PSO severe |
|
palmo-plantar both hands or both soles severity 4 for erythema, induration, scaling, fissuring |
PASI = 8 → PSO moderate or ESIF = 24 → PSO severe |
Abbreviations: PASI - Psoriasis Area and Severity Index; BSA - Body Surface Area; PSSI - Psoriasis Scalp Severity Index; NAPSI - Nail Psoriasis Severity Index; ESIF - Erythema, Scaling, Induration, Fissuring.
In a series of 26 psoriasis patients having nail psoriasis [33], there was a significant moderate positive correlation between the NAPSI score and DLQI (p=0.001). A me-ta-analysis published in 2019 showed that the prevalence of palmoplantar psoriasis (PPP) of both palms and soles (59%) was approximately three times higher than the prevalence of any single location of PPP, either palms (21%) or soles (20%). More than 60% of the patients from the 15 studies included in this meta-analysis had PPP with at least one ad-ditional area involved [34,35].
The severity assessment in patients with several affected areas and at least one dif-ficult-to-treat area can also raise issues from the point of view of insurance and health authorities: the existence of country-specific therapeutic protocols limits the use of sys-temic therapy (biologics, small molecules) for severe psoriasis only to be done according to the PASI/PGA/BSA/DLQI scores. To meet psoriasis patients’ needs while respecting the requirements of the authorities involved, we propose to be taken into account the score that represents the highest degree of severity.
The possible solutions for such practical issues reside in making the classification by degree of severity according to the highest calculated score, regardless of whether it is PASI or PSSI, NAPSI, ESIF. If for the 3 examples mentioned above (affecting at least 2 areas, one of which being difficult-to-treat), the severity assessment would be done ac-cording to the specific scores of the special areas, then the patients would fall into the category of severe psoriasis, with all of the implications deriving from this classification (choosing the right treatment according to the involvement of a difficult-to-treat area, setting the therapeutic goal, evaluating the success or failure of the therapy). The challenge of this new possible classification could be the subsequent monitoring of the effectiveness of the treatment.
Since PASI is considered as one of the standards in clinical trials (allowing historical comparison with several treatments, having a good correlation with other objective out-come measures, being the most validated objective measurement of psoriasis severity, with a test-retest variability of less than 2%), another solution could be to add a correction factor in the calculation of PASI for special areas, as in the example of the first patient with 2 areas involved, scalp and trunk (Table 3). The introduction of a correction factor for that area of the body defined as hard-to-treat could change the importance of the scalp involvement in the calculation of PASI and consequently the patient's framing in the severity degrees, reflecting the reality of usual clinical practice.”
We have also made changes in the Discussion part of the manuscript: “Psoriasis of the scalp, face, intertriginous areas, genitals, hands, feet, and nails is of-ten underdiagnosed, and under-treated. In spite of the small surface area which is com-monly affected by psoriatic lesions in these locations, patients have disproportionate levels of physical impairment and emotional distress. Limitations in current disease se-verity scores do not fully assess the impact of disease on a patient’s quality of life, many patients not receiving adequate care. In these cases, the therapeutic attitude to adapt is to adapt therapy (dose increase, therapeutic combination), change of treatment (switch) or continuation of treatment for the next 3 months with reevaluation [59]. Psoriasis severity classification is a major problem which needs to be addressed in order to guide the phy-sician’s decisions regarding treatment, having always in mind that the disease is hetero-geneous in clinical expression and response to treatment, in its duration and involved areas (including percentage of body area), being constantly variable [22].
Psoriasis is highly influenced by external factors; the list of triggering factors for flare-ups is extensive, starting with stress, mild localized skin trauma, different infections and drugs for different co-morbidities, alcohol consumption, smoking, weather etc. Seventy three percent of the patients have at least one comorbidity that can influence the disease evolution and response to treatment, especially for difficult-to-treat areas [59]. In the given example – the patient with 2 areas affected by psoriasis has been evaluated after 3 months of treatment with ∆ PASI 90 for lesions on the trunk and ∆ PASI < 50 for scalp. The patient suffered a great level of stress during these 3 months but according to DLQI there was significant improvement.
There are many systems classifying psoriasis severity, but none have managed to reach a consensus, not having clear-cut demarcation between severity degrees (moderate and severe), the methodology disregarding the involvement of psoriasis in difficult-to-treat areas; the fact that current practices employ systems that are mainly used by physicians/dermatologists (such as PASI, PGA and/or BSI) and by patients (DLQI) alike is a step further in psoriasis severity assessment and treatment, the quality of life being of paramount importance. [22]. Figure 2 reveals the raised issues, with point-by-point ex-planations and proposed solutions for the clinicians in their daily practice.
Figure 2. Issues and possible solutions in psoriasis severity effective calculation.
Abbreviations: PASI - Psoriasis Area and Severity Index; BSA - Body Surface Area; PSSI - Psoriasis Scalp Severity Index; NAPSI - Nail Psoriasis Severity Index; ESIF - Erythema, Scaling, Induration, Fissuring.
For psoriasis involving difficult-to-treat areas the challenge extends from establishing severity and choosing the optimal treatment to correctly evaluating the efficacy or failure of the therapy; this is a major problem which our paper addressed, having the advantage of bringing forth a practical, clinical issue which dermatologists face. There are several solutions to this issue; the first one is the evaluation of different scores for each area and the success or failure of the treatment to be evaluated by the area with the smallest improvement. The second solution is the inclusion of special areas in the treatment goal algorithm or to use a completely different algorithm for psoriasis in difficult-to-treat areas by using the data obtained in the different clinical studies or real-world experience. The current paper has made a summary of the literature at hand, examining the current knowledge on psoriasis and its score assessment, with the limitation of possibly having a broad area of research data (a wide range of psoriasis score reporting analysis), but with the advantage of identifying possible new research areas.”
Response for Reviewer 4: Thank you for your suggestions. We added and included in the text all your suggested changes and necessary information.
Thank you for all your consideration!

Round 2
Reviewer 3 Report
The revised article is well done, with the revisions made it has definitely improved. It can be accepted in this version.